# A Review of Antifungal Susceptibility Testing for Dermatophyte Fungi and It’s Correlation with Previous Exposure and Clinical Responses

**DOI:** 10.3390/jof8121290

**Published:** 2022-12-09

**Authors:** Sidra Saleem Khan, Roderick James Hay, Ditte Marie Lindhardt Saunte

**Affiliations:** 1The Dermatology Centre, Salford Royal NHS Foundation Trust, Manchester M6 8HD, UK; 2St. John’s Institute of Dermatology, King’s College London, London SE1 9RT, UK; 3Department of Dermatology, Zealand University Hospital, DK-4000 Roskilde, Denmark; 4Institute of Clinical Medicine, Faculty of Health Science, University of Copenhagen, DK-1350 Copenhagen, Denmark

**Keywords:** dermatophyte antifungal resistance

## Abstract

**Background:** An increase in the number of recurrent and recalcitrant dermatophytoses calls for a tool to guide the clinician to correlate in vitro minimum inhibitory concentration (MIC) data, antifungal treatment with clinical outcomes. This systematic review aims to explore a possible correlation between one aspect of this, previous antifungal exposure, and clinical outcomes. **Methods:** A systematic literature search for articles on previous antifungal treatment, treatment outcome, susceptibility methods used, organism (genus/species), and MIC values was conducted. **Results:** A total of 720 records were identified of which 19 articles met the inclusion criteria. Forty percent of the cases had contact with or travel to India, 28% originated from or had traveled to other countries where treatment unresponsive tinea infections had been reported. Tinea corporis was the most common clinical presentation and the species involved were *Trichophyton* (*T.*) *indotineae* and *T. rubrum*, followed by *T. mentagrophyte*/*interdigitale* complex and *T. tonsurans*. Nearly all patients had previously been exposed to one or more antifungals. The studies were too heterogeneous to perform a statistical analysis to test if previous antifungal exposure was related to resistance. **Conclusions:** Only a few studies were identified, which had both sufficient and robust data on in vitro susceptibility testing and clinical treatment failure. Further research on the value of susceptibility testing to improve clinical practice in the management of dermatophyte infections is needed.

## 1. Introduction

Dermatomycoses, otherwise known as fungal infections of the skin, and structures derived from it such as hair and nails are very common globally and carry a significant associated burden of disease in terms of prevalence and years lived with disability [1]. Fungal infections of the skin are also known to have an impact on psychosocial wellbeing, with affected individuals reporting lower self-esteem, social withdrawal and embarrassment as a result of their infection [2,3]. The common dermatomycoses are dermatophytosis, *Malassezia* infections and cutaneous candidiasis.

Recent years have not only seen a rise in the number of cases of dermatophytosis in some parts of the world but also, alarmingly, an increase in the number of recurrent and recalcitrant cases [4]. These include infections that, previously, would have been considered to be responsive to oral antifungals given in conventional doses and treatment durations, such as tinea corporis and tinea cruris. There are a number of reasons that may explain treatment failure including poor compliance with treatment, variability in the quality of antifungal drugs including generic products, misuse of strong topical corticosteroids or steroid combinations and host immune dysfunction [5]. Although these are important features which partially explain transmission and spread into new populations, resistance by dermatophytes to the commonly used antifungals is increasingly being reported globally and is also being cited as a further reason behind the recent rise in recurrent or recalcitrant fungal skin disease [5].

Terbinafine is a fungicidal drug working by inhibiting the squalene epoxidase (SQLE) enzyme thereby interfering with the ergosterol synthesis, which is a component of the fungal cell membrane [6]. It is widely used as the first line agent against many dermatophyte species. The growing threat of terbinafine resistance, particularly amongst members of the *Trichophyton* (*T*.) *mentagrophytes-interdigitale* complex, including the newly identified *T. indotineae* on the Indian subcontinent has been a cause for concern for many [4]. Point mutations in the squalene epoxidase (SE) gene have been identified in such species leading to substitutions of amino acid positions (F397L, L393F, L393S, H440Y, F393I, F393V, F415I, F415S, F415V, S443P, A448T, L335F/A448T, S395P/A448T, L393S/A448T, Q408L/A448T, F397L/A448T, F397I, L437P, I121M/V237I and H440Y/F484Y, Y414C/L438C) [6,7,8,9,10], most of which are associated with reduction of antifungal sensitivity, but this information on its own does not provide a specific minimum inhibitory concentration (MIC) value. Antifungal susceptibility testing (AFST) is used to determine the MIC of a given drug with the purpose of predicting if a patient will respond to standard antifungal therapy. Breakpoints are used to categorize MIC results into three susceptibility categories: (S) Susceptible where standard dosing regimen has a high likelihood of therapeutic success, (I) Susceptible but requires increased exposure to the antifungal agent by adjusting the dosing regimen or by its concentration at the site of infection and (R) Resistant where there is a high likelihood of therapeutic failure even when there is increased exposure [11]. Epidemiological cutoff values (ECVs or ECOFFs) are used, when there is no breakpoint available. ECVs are MIC thresholds for a given antifungal, which are used to discriminate between wild-type (WT) which is defined a strain without any phenotypically expressed resistance mechanism and non-WT strains of a specific species, and they are used to identify isolates that may have acquired resistance to a particular antifungal and thus may be less likely to respond to therapy [12].

Comparing AFST of dermatophytes is challenging because various methods have been used to determine MIC: E-test, agar dilution, agar disc diffusion, macro- and microbroth dilution methods [13]. The AFST methods differ in inoculum concentration, incubation temperature, incubation time, different culture media and end-point criterion of fungal growth, which makes it difficult to compare results across studies. E-test is based on the use of a strip incorporating antifungal concentrations ranging from low to high. The inoculum suspensions is prepared from dermatophyte colonies, and the concentration (CFU/mL) can be adjusted using a turbidimeter and the E-test strips is placed in the center of the growth media and incubated [14]. The drug concentration shown on the E-test strip at the outer border of the elliptical inhibition halo was recorded as MIC [14]. When using an agar dilution method, the antifungal agent is added to agar plates in different dilutions (concentrations). The pathogen to be tested is added to each plate and incubated at a specific temperature and incubation time. MIC is the lowest concentration of antifungal that inhibit the growth of the dermatophyte [15]. Agar disc diffusion methods use discs containing a fixed concentration of antifungal agent are placed on the inoculated agar surface. After a specific incubation time at fixed temperature the zones of growth inhibition around each of the antifungal discs are measured and compared with a reference strain of known specific MIC [15]. Macro- and microbroth dilution tests use liquid growth medium dispensed in test tubes with twofold dilutions of antimicrobial agent under test (e.g., 1, 2, 4, 8 and 16 ug/mL). The tubes are inoculated with standardised inocula and incubated at a specific temperature. MIC is determined as the lowest concentration of antimicrobial agent that prevents growth or visible growth as evidenced by turbidity [15]. When disposable plastic plates with small wells are used the method is called ‘microbroth dilution test’.

AFST regarded as a specialized procedure only available in a few laboratories. Dermatophyte or ringworm infections are therefore commonly treated without recourse to antifungal susceptibility testing, whether in general or dermatological medical practice. This has been custom and practice over many years despite the fact that failure of antifungal treatment is encountered regularly, particularly in the management of onychomycosis, where treatment failure is usually linked to other factors such as poor penetration of the drug into the infected nail plate, poor compliance or altered drug metabolism due to concurrent therapies. Only in comparatively few studies has the in vitro sensitivity of dermatophytes been investigated in the assessment of poor treatment responses. However, a few such studies are available and, as more recently cases of antifungal resistance confirmed by in vitro testing have been reported, the demand for such tests has increased, for instance with the wider spread of *T. indotineae* infections in India and elsewhere [7,8]. For this reason, the need for laboratory assessment of in vitro resistance, previously in low demand, is now increasing. As an example, a recent pan European survey of clinical unresponsive dermatophyte infections showed that only a half of fungi cultures from lesions had been tested for antifungal susceptibility [16]. These observations also exclude the significant numbers of cases where no cultures had been taken.

Furthermore, AFST of dermatophytes remains poorly standardized and there is a lack of consistency within guidelines correlating in vitro MIC data with clinical outcomes.

This finding has triggered the wider consideration of susceptibility testing resulting in this systematic review of AFST and it’s correlation with previous exposure and clinical outcomes. This is a prerequisite for understanding the extent of the problem and establishing measures to address it.

## 2. Material and Methods

The review was conducted according to the Preferred Reporting Items for Systematic Reviews and Meta- Analysis (PRISMA)-guidelines [17]. These guidelines provide a minimum set of evidence based items that require reporting on as part of a checklist when conducting systematic reviews. On the 31 May 2022 PubMed was searched. For specified search strings please see Appendix A.

The inclusion criteria were studies with patient data on previous oral treatment and treatment outcome, name of susceptibility method, organism (genus/species), and MIC value. The following data were collected if available: the patient’s immune status, infection duration, anatomical area of infection, MIC value (incl. MIC50, MIC90, range), and mutation detected. Articles excluded were those presenting data on other specific types of dermatophytoses such as tinea capitis, onychomycosis and studies in languages other than English, as well as reviews and meta-analyses.

Two researchers (SSK and DMLS) independently screened titles and abstracts for eligibility using an online software tool, known as Rayyan, which allow a blinded screening of title, abstracts and full-text papers blinded [18,19]. All relevant articles were full text assessed by SSK or DMLS, and texts were discussed with before inclusion and if in doubt of eligibility with RJH. The literature lists of the included articles were checked for further relevant literature.

## 3. Results

### Literature Search

A total of 720 records were identified through database searching from which 153 duplicates were removed. In total 567 articles were screened, 24 were full text assessed for eligibility, and 19 articles met the inclusion criteria (Figure 1). For the final list of included articles please see Appendix B.

In total there were 75 patients reported in these studies across the 19 papers included within this review. The mean age of this set of patients was 39.2 years (*n* = 53, range 4–81 years). There were 37 males and 17 females (21 undocumented gender). Forty percent (*n* = 30/75) of the patients were noted to have had contact with or travel to India, 28% (*n* = 21/75) did not and in 32% (*n* = 24/75) there was no information provided on travel. A total of 28% (*n* = 21/75) of the patients originated from or had traveled to other countries which included Bangladesh (*n* = 9/21), Iran (*n* = 6/21), Japan (*n* = 2/21), Myanmar (*n* = 1/21), Nepal (*n* = 1/21), Yemen (*n* = 1/21) and Sri Lanka (*n* = 1/21). Some of these countries have been considered to be potentially affected in published reports of the growing spread of recalcitrant tinea infections because clinical problems with treatment failure [9]. Only six patients were noted to have other co-morbidities which included diabetes mellitus (*n* = 3), Cushing’s syndrome (*n* = 1), Darier disease (*n* = 1) and congenital ichthyosiform erythroderma (*n* = 1), whereas 19 patients were noted to have no pre-existing co-morbidities and for the remaining 50 patients there was no mention of their past medical histories.

The clinical presentation of the infection was available in 88% (*n* = 66/75) of the patients. Tinea corporis was noted to be the most common presentation/anatomical site involved (80.3%, *n* = 53/66), followed by tinea cruris (24.2%, *n* = 16/66), tinea pedis (21.2%, *n* = 14/66), legs (22.3%, *n* = 15/66), hands (15.2%, *n* = 10/66), perineum (15.2%, *n* = 10/66), tinea facei (9.1%, *n* = 6/66), arms (9.1%, *n* = 6/66), and external genitalia (4.5%, *n* = 3/66).Where documented, 40 patients had more than one anatomical site involved or different types of clinical presentation.

The most prevalent dermatophytes species observed were *T. indotineae* (38.7%, *n* = 29/75) and *T. rubrum* (38.7%, *n* = 29/75), followed by dermatophytes of the *T. mentagrophyte/interdigitale* complex (20%, *n* = 15/75), and *T. tonsurans* (2.7%, *n* = 2/75). Details on patient demographics, dermatophyte species and SQLE mutation is available in Table 1.

A total of 5 studies used the European Committee for Antimicrobial Susceptibility Testing (EUCAST) method (18 isolates/patients in total) (Table 2), 11 studies used the Clinical and Laboratory Standards Institute (CLSI) method (33 isolates/patients in total) (Table 3) and 1 study used the NCCLS method, which is identical to the CSLI version (1 isolate/patient). There were two studies (24 isolates/patients in total) that did not specify what method of AFST they used. One study used a specifically designed assay for griseofulvin sensitivity (Table 4)

Almost all patients (97.3%, *n* = 73/75) had previous exposure to one or more antifungals. Of the antifungal therapies used previously, terbinafine was the most common (61.3%, *n* = 46/75), followed by griseofulvin (32%, *n* = 24/75), fluconazole (9.3%, *n* = 7/75), ketoconazole (6.7%, *n* = 5/75), clotrimazole (5.3%, *n* = 4/75), itraconazole (4%, *n* = 3/75), sertaconazole (4%, *n* = 3/75), bifonazole (4%, *n* = 3/75), miconazole (2.7%, *n* = 2/75), ciclopirox (2.7%, *n* = 2/75), luliconazole (2.7%, *n* = 2/75), ravuconazole (1.3%, *n* = 1/75) and omoconazole (1.3%, *n* = 1/75). There were 29 patients, who were initially noted to have had “no response” or “minimal response” to previous antifungal therapy, and then eventually were reported to have “cleared” or been “cured” of their dermatophyte infection.

The MIC’s in patients previously treated with terbinafine was >1 mg/L in 82% (9/11) for *T. indotineae* and <1 mg/L in 100% (5/5) of terbinafine naïve cases which had highest MIC levels of <0.06 mg/L using the EUCAST method (Table 2). Using the CLSI methods only one of twelve patients infected with *T. indotineae* and previously treated with terbinafine had a low MIC 0.125 mg/L and eleven of twelve had a higher MIC > 1 mg/L (Table 3). It is difficult to comment on the practical implications of this finding as it is likely that dermatophytes isolated from patients who had not responded would have been tested in preference to the responders. However, because of the higher number of cases of sensitive strains amongst those who had not received terbinafine it remains a strong possibility that treatment with terbinafine selects for the development of strains with squalene epoxidase mutations. Further studies on pretreatment isolates would be important. None of the patients were exposed to itraconazole or griseofulvin before the AFST (Table 2 and Table 3).

The most common antifungal therapy used after AFST, in these patients, to achieve “clearance” or “cure” was itraconazole (*n* = 22/29), followed by griseofulvin (*n* = 5/29), terbinafine (*n* = 2/29), clotrimazole (*n* = 2/29), voriconazole (*n* = 2/29), eberconazole (*n* = 2/29), sertaconazole (*n* = 1/29), fosravuconazole (*n* = 1/29) and topical luliconazole (*n* = 1/29).

Due to the heterogeneity of the studies (few cases, different methods, etc.) it was not possible to perform a statistical analysis of these results to test if previous exposure to an antifungal was related to antifungal resistance.

## 4. Discussion

This study establishes two key points. Firstly, there have been very few studies of clinical antifungal resistance in dermatophyte infection, which have been backed by in vitro data, thereby allowing any correlation between failure of treatment with drug resistance. Secondly, it is clear that there are specific strains of dermatophyte fungi that are resistant to the commonly used antifungal agents, such as the *T. indotineae* isolates, associated with the outbreak of dermatophytosis first reported in India.

These findings need to be qualified by the fact that the study did not include infections of the nails, onychomycosis, or the scalp, tinea capitis, both of which, but particularly the former, have been associated with treatment failure. However, failure of nail infections to respond clinically to appropriate antifungals has been associated with factors other than antimicrobial resistance [20]. These include poor drug absorption or enhanced antifungal elimination due to concomitant administration of other drugs that enhance metabolism, poor compliance with treatment because of its long duration as well as abnormalities in the anatomy of the infected nail plate that leads to reduced and subinhibitory drug concentrations in some areas such as the longitudinal nail streaks, known as dermatophytomas. Our study focused on tinea corporis and cruris that are sites of infection where there are fewer obstacles to the “normal” distribution of antifungal drugs at the site of infection, although there are some exceptions such as patients with certain keratodermas who often respond poorly to antifungals [21] as well as non-compliance or side effects. The finding of in vitro resistance, now associated with outbreaks of infection, raise concerns about the ease with which resistant strains can spread in and across community boundaries and whether MIC determinations should be more widely adopted as a standard clinical practice in dermatology where this is feasible. In addition, we need strategies to treat these infections. The Indian Association of Dermatology, Venereology and Leprosy (IADVL) has proposed modifications to the treatment which are particular apposite for the isolates associated with the outbreak in the India subcontinent [22], where terbinafine resistance often is associated with mutations on the squalene epoxidase gene, have been identified regularly. Their recommendations include longer and higher doses of itraconazole, a strategy which has been deployed with good results; more studies assessing these changes are still needed. However, in addition some of the earlier studies reported here occurred long before the *T. indotineae* outbreak first emerged [23]. Treatment unresponsive dermatophytosis presenting as persistent or recurrent tinea corporis or cruris, usually caused by *T. rubrum*, has been known for many years. Unfortunately, few of these isolates were tested for in vitro resistance apart from those reported in the Artis study [24]. So although clinically unresponsive (clinically resistant) cases have been noted for many years [25], they have just not been investigated fully including the drug sensitivity of the fungi isolated. The patients in this series had few underlying comorbidities although other abnormalities such as diabetes mellitus, Cushing’s syndrome, Darier disease and congenital ichthyosiform erythroderma were recorded in six cases [26,27,28,29]. Other comorbidities that have been described previously as associations with persistent tinea, but not cited in the publications included here, have been chronic mucocutaneous candidiasis and palmoplantar keratodermas as well as deep dermatophytosis associated with CARD 9 mutations [30].

Interpretation of in vitro sensitivity tests such as MICs, applied to dermatophytosis, also needs to be clarified. There is a rule of thumb, which states that infections due to susceptible strains respond to appropriate therapy in 90% of cases, whereas infections due to resistant strains respond in approximately 60% of patients [31]. However, as this was based mainly on data from systemic fungal infections it is not clear to what extent it is applicable to dermatophytosis. The benefit of susceptibility testing has to be of practical value in the management of patients and it is appropriate to ask whether the categorization of an isolate as either susceptible or resistant will help to predict the patient’s response to therapy and guide clinicians to adopt measures that provide successful outcomes. Guidelines that enunciate clearly the indications for in vitro sensitivity testing in dermatophyte infections need to be drawn up. EUCAST has recently published tentative ECOFF values for *T. indotinea* (itraconazole 0.25 mg/L, terbinafine 0.125 mg/L, voriconazole 1.0 mg/L and amoralfin 0.5 mg/L) and for *T. rubrum* (itraconazole 0.25 ug/mL, terbinafine 0.03 mg/L, voriconazole 0.125 mg/L and amorolfine 0.125 mg/L) [32]. ECOFF do not predict clinical response, but identify non-wild type (WT) or less susceptible isolates and our data confirm that 82% (9/11) of the previously terbinafine exposed isolates were non-wild type, and 100% (2/2) of the *T. rubrum* isolates, respectively. CLSI suggest that isolates with terbinafine MIC up towards 0.25 mg/L is without resistance mechanisms and clinically susceptible, but this value is however several fold higher than recommended CLSI MIC range for the *T. mentagrophytes* quality control strain (0.002–0.008 mg/L) which may raise the concern if 0.25 mg/L is perhaps a too high.

The spread of *T. indotineae* infections in India and now to other countries including those further afield, eg in Europe, has been a wakeup call for action. It has simultaneously raised questions about ease of spread of infection in communities, the role of ancillary factors such as abuse of corticosteroid therapies and the need for new diagnostic methods and antifungal treatment strategies [33]. With *T. indotineae* infections there seems to be a clearer relation between in vitro results and in vivo responses, but the position of other infections such as the *T. rubrum* cases reported by Artis and others are less clear [24]. These cases were not “one off” examples as similar cases have been reported by others regularly over many years.

This is, therefore an area that now needs some urgent answers and the starting point is further research on the value of susceptibility testing in improving clinical practice in the management of dermatophyte infections. Recently the World Health Organisation (WHO) has issued a report on fungal pathogen priorities focusing on systemic pathogens [34]. In its conclusion it highlights the need for better diagnostics, more frequent and more accessible antifungal susceptibility testing and the need to include surveillance for fungal resistance as a public health priority. These conclusions are equally applicable to the rising tide of dermatophyte resistance. The authors of the WHO report suggest that subsequent work on priorities amongst pathogenic fungi might focus on a wider group of organisms including superficial mycoses. Our study suggests that such work should not be unduly delayed.

## Figures and Tables

**Figure 1 jof-08-01290-f001:**
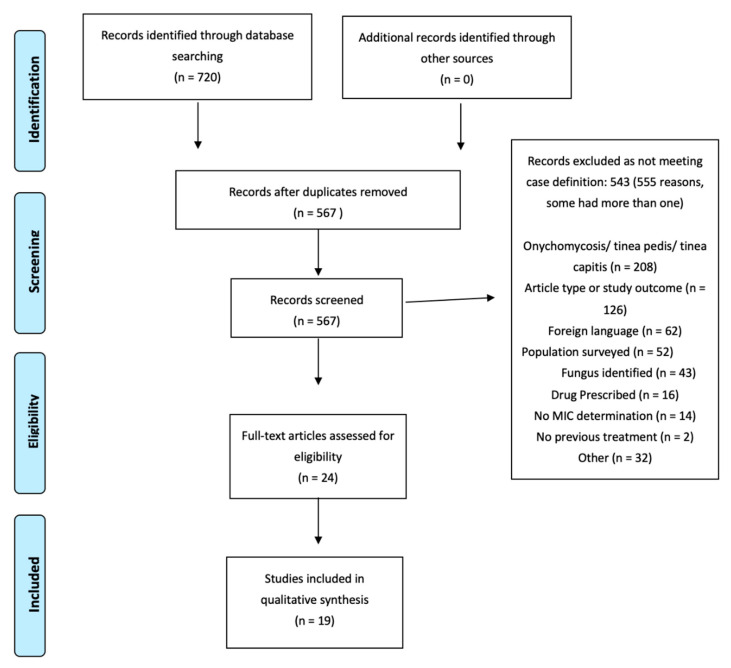
PRISMA Flow Diagram, Flowchart of literature search.

**Table 1 jof-08-01290-t001:** Information on isolates, clinical characteristics of patients, contact with endemic areas, mutations of treatment resistant cases.

Total (*n* = 75)	Gender F/M/NA	Mean Age in Years (Range)	Contact with India/Asia *n*/Total (%)	Contact with Middle East	Disease Duration in Months (Range) or Chronic	Mutation
*T. indotineae* (*n* = 9)	NA	NA	9/9 (100) Bangladesh (*n* = 6) Myanmar (*n* = 1 Japan (*n* = 2)		NA (*n* = 9)	L393S (*n* = 2) F397L (*n* = 4) A448T (*n* = 2) No mutation (*n* = 1)
*T. mentagrophytes/interdigitale* (*n* = 4)	1/1/2	24.5 (22–27) NA (*n* = 2)	4/4 (100) India (*n* = 3) Nepal & India (*n* = 1)		15.5 (7–24) Chronic (*n* = 0) NA (*n* = 2)	F397L (*n* = 4)
*T. mentagrophytes* ITS type VIII (*n* = 20)	11/9/0	39.4 (4–64)	India (*n* = 15) Bangladesh (*n* = 3) Sri Lanka (*n* = 1)	Iran (*n* = 4) Yemen (*n* = 1)	7.9 (3–12) Chronic (*n* = 0) NA (*n* = 11)	F397L (*n* = 11) A448T (*n* = 2) c.1342G > A in the SQLE (*n* = 2) L393S (*n* = 1) *n* A1223T (*n* = 2) No mutation (*n* = 2) Two mutations (*n* = 1)
*T. interdigitale* (*n* = 11)	0/1/10	47 (–) NA (*n* = 10)	India (*n* = 11)		NA (*n* = 11)	F397L (*n* = 7) L393F (*n* = 2) No mutation (*n* = 2)
*T. rubrum* (*n* = 29)	5/24/0	40.8 (9–81) NA (*n* = 1)	Japan (*n* = 1)		65.0 (24–138) Chronic (*n* = 25) NA (*n* = 0)	
*T. tonsurans* (*n* = 2)	0/2/0	25.5 (25–26) NA (*n* = 0)		Iran (*n* = 2)	12(–) Chronic (*n* = 0) NA (*n* = 1)	

F: female, M: male, *n*: number, NA: not available, *T.: Trichophyton.*

**Table 2 jof-08-01290-t002:** MIC values using European Committee for Antimicrobial Susceptibility Testing method according to previous antifungal treatment.

	Terbinafine											
Species	MIC Value mg/L	0.016	0.03	0.06	0.125	0.25	0.5	1	2	4	8	>8
*T. indotineae* (16)	Yes			2				1	3	4		1
	No	1	1	3								
*T. rubrum* (2)	Yes					1				1		
	No											
	**Itraconazole**											
**Species (*n*)**	**MIC value** **mg/L**	**0.016**	**0.03**	**0.06**	**0.125**	**0.25**	**0.5**	**1**	**2**	**4**	**8**	**>8**
*T. indotineae* (14)	Yes			1								
	No	3		7	2	1						
*T. rubrum* (2)	Yes											
	No		1		1							

MIC: minimum inhibitory concentration; *T.: Trichophyton*; menta: mentagrophytes; inter: interdigitale.

**Table 3 jof-08-01290-t003:** MIC values using Clinical Laboratory Standard Institute susceptibility method according to previous antifungal treatment.

	Terbinafine															
Species	MIC Value mg/L	0.015	0.0625	0.03	0.06	0.125	0.25	0.5	1	>1	2	4	8	>8	16	≥32
*T. indotineae*	Yes					1				2	2	1	1	3		2
	No													1		
*T. menta/inter* complex	Yes							2			2		1		1	9
	No															
*T. rubrum*	Yes															3
	No															
*T. tonsurans*	Yes											2				
	No															
	**Itraconazole**															
**Species**	**MIC value** **mg/L**	**0.015**	**0.0625**	**0.03**	**0.06**	**0.125**	**0.25**	**0.5**	**1**	**>1**	**2**	**4**	**8**	**>8**	**16**	**≥32**
*T. indotineae*	Yes															
	No	2		4		1		1			1	3			1	
*T. menta/inter* complex	Yes															
	No		2	2												
*T. rubrum*	Yes															
	No			1		1		1							1	
*T. tonsurans*	Yes															
	No								2							
	**Griseofulvin**															
**Species**	**MIC value** **mg/L**	**0.015**	**0.0625**	**0.03**	**0.06**	**0.125**	**0.25**	**0.5**	**1**	**>1**	**2**	**4**	**8**	**>8**	**16**	**≥32**
*T. menta/inter* complex	Yes															
	No				1	3		1	1							
*T. rubrum*	Yes															
	No						1						1			

MIC: minimum inhibitory concentration; *T.: Trichophyton;* menta: mentagrophytes; inter: interdigitale.

**Table 4 jof-08-01290-t004:** MIC values using an inhouse susceptibility testing for griseofulvin resistant *T. rubrum*.

MIC Value mg/L	0.5	1	2	3	4	5	6	9	>18
**Griseofulvin**									
Yes	1	4	2	3	3	3	2	1	2
No									

MIC: minimum inhibitory concentration; *T.: Trichophyton.*

## Data Availability

Data is contained within the article or Appendix A and Appendix B.

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
