# Peer review of "A Review of Antifungal Susceptibility Testing for Dermatophyte Fungi and It’s Correlation with Previous Exposure and Clinical Responses"

_jof, 2022, doi:10.3390/jof8121290_

Round 1

Reviewer 1 Report

Dear Authors,

This is a very interesting and comprehensive work really i enjoyed reading it please see my comments in the attached file and this work is really important especially when the WHO recently declared the top fungal pathogens list and we need all to improve as well as to detect the limitation of the up-to-date knowledge which in turn will help us to fight against fungal infections including dermatophytes which considered as a public health issue 

Author Response

Title: please add here dermatophytes. For the first time i guess this article for all human pathogenic fungi please specify

Thank you for your suggestions. We have now changed the title to: “A review of antifungal susceptibility testing for dermatophyte fungi and it’s correlation with previous exposure and clinical responses in humans”.

Add in brief terbinafine mechanism of action please just a few words

Terbinafine is a fungicidal drug working by inhibiting the squalene epoxidase (SQLE) enzyme thereby interfering with the ergosterol synthesis which is a component of the fungal cell membrane. It is widely used as the first line agent against many dermatophyte species. Please see line 65-67 (Ref 6).

Please elaborate more, and whether they correlated the different SNPs with the MIC and which SNPs associated with low MIC and so on

Add a reference (page 2 line 83)

Font size please check

Many thanks for noticing this, this has now been reveiwed and corrected at line 247 to the font Palatino, size 10.

Font size at line 247

The font size at line 247 has been changed to size 10.  

Reviewer 2 Report

There are some suggestions to improve the quality of the paper:

1. The manuscript contains many typo and format errors, like line 247, table captions and others.

2. The title should be more specific regarding which fungal genus and disease have been considered for the study.

3. In abstract, add the full genus names of fungus.

4. Lines 48-51 needs to rephrase for better understanding.

5. Include and discuss the reference PLoS Pathog. 2022 Sep 29;18(9):e1010795. doi: 10.1371/journal.ppat.1010795.

6. In the introduction section discussing the details of different MIC determining methods seems to be unnecessary, better to discuss in details about more relevant information such as the mutations and resistance leading to treatment failure.

7. Include reference for the line 100, 111, lines 237-240, 257-264.

8. Full form of all abbreviations should be mentioned at the first mention, following which only the abbreviations should be used.

9. Discuss in brief about PRISMA guidelines and the online software tool in methodology.

10. Supplementary table 1 is not attached.

11. MIC values needs to be expressed with proper units.

12. There is no mention of table 3 and 4 in results section, its needs to be discussed in details what they represent.

13. In Discussion section the 2 key points are not clear (lines 202-206).

14. In Figure 1, what does wrong population, fungus and drug means?

Author Response

There are some suggestions to improve the quality of the paper:

  1. The manuscript contains many typo and format errors, like line 247, table captions and others.

Many thanks for pointing this out. The paper has been re-read by all authors and corrections have been made accordingly.

  1. The title should be more specific regarding which fungal genus and disease have been considered for the study.

Thank you for your suggestions. We have now changed the title to “A review of antifungal susceptibility testing for dermatophyte fungi and it’s correlation with previous exposure and clinical responses in humans”.

  1. In abstract, add the full genus names of fungus.

Thank you for noticing this. It has now been changed accordingly within the abstract.

  1. Lines 48-51 needs to rephrase for better understanding.

This sentence at lines 48-51 has now been removed.

  1. Include and discuss the reference PLoS Pathog. 2022 Sep 29;18(9):e1010795. doi: 10.1371/journal.ppat.1010795.

Many thanks for this recommendation. This reference has now discussed and added (see Ref 33).

  1. In the introduction section discussing the details of different MIC determining methods seems to be unnecessary, better to discuss in details about more relevant information such as the mutations and resistance leading to treatment failure.

We agree, but this section has been added as Journal of Fungi wanted the manuscript to be longer. Additionally, we have discussed mutations on page 2, lines 92- 97.

‘Point mutations in the squalene epoxidase (SE) gene have been identified in such species leading to substitutions of amino acid positions (F397L, L393F, L393S, H440Y, F393I, F393V, F415I, F415S, F415V, S443P, A448T, L335F/A448T, S395P/A448T, L393S/ A448T, Q408L/A448T, F397L/A448T, F397I, L437P, I121M/V237I and H440Y/F484Y, Y414C/L438C)6,7,8,9,10, most of which are associated with reduction of antifungal sensitivity, but this information on its own does not provide a specific minimum inhibitory concentration (MIC) value.’

  1. Include reference for the line 100, 111, lines 237-240, 257-264.

Many thanks for noticing this. All references have now been added to those points/lines.

  1. Full form of all abbreviations should be mentioned at the first mention, following which only the abbreviations should be used.

Many thanks for noticing this. The authors have re-reviewed the paper and made the appropriate changes.

  1. Discuss in brief about PRISMA guidelines and the online software tool in methodology.

We have now included a brief sentence with regards to PRISMA and the name of the software used. It now reads:

Page 3 lines 201-203:

“The review was conducted according to the Preferred Reporting Items for Systematic Reviews and Meta- Analysis (PRISMA)-guidelines17. These guidelines provide a minimum set of evidence based items that require reporting on as part of a checklist when conducting systematic reviews.“

 Page 3 lines 214-215:

“Two researchers (SK and DMS) independently screened titles and abstracts for eligibility using an  online software tool, known as Rayyan, which allow a blinded screening of title, abstracts and full-text papers blinded 18,19. All relevant articles were full text assessed by SK or DMS, and texts were discussed with before inclusion and if in doubt of eligibility with RJH. The literature lists of the included articles were checked for further relevant literature.”

  1. Supplementary table 1 is not attached.

This is now included as an appendices (appendix 1).

  1. MIC values needs to be expressed with proper units.

Thank you for noticing this. We have added ‘mg/L’ after the MIC values.

  1. There is no mention of table 3 and 4 in results section, its needs to be discussed in details what they represent.

Thank you for this comment. You are right.  We have added following:

Page 4, lines 285-290:

 A total of 5 studies used the European Committee for Antimicrobial Susceptibility Testing (EUCAST) method (18 isolates/patients in total) (Table 2), 11 studies used the Clinical and Laboratory Standards Institute (CLSI) method (33 isolates/ patients in total) (Table 3) and 1 study used the NCCLS method, which is identical to the CSLI version (1 isolate/patient). There were two studies (24 isolates/patients in total) that did not specify what method of AFST they used. One study used a specifically designed assay for griseofulvin sensitivity (Table 4)

And at page 4, lines 301-306:

The MIC’s in patients previously treated with terbinafine was >1 mg/L in 82% (9/11) for T. indotineae and < 1 mg/L in 100% (5/5) of terbinafine naïve cases which had highest MIC levels of < 0.06 mg/L using the EUCAST method (Table 2). Using the CLSI methods only one of twelve patients infected with T. indotineae and previously treated with terbinafine had a low MIC 0.125 mg/L and eleven of twelve had a higher MIC >1 mg/L (Table 3).

And at page 4, lines 311-312:

None of the patients were exposed to itraconazole or griseofulvin before the AFST (Table 2& 3).    

This was an omission and the references to these have now been inserted in the results section at the relevant section where it is discussed at line 287-288.

  1. In Discussion section the 2 key points are not clear (lines 202-206).

Thank you. We have changed it and hope that this is clearer

From page 5, lines 379-384.

This study establishes two key points. Firstly, there have been very few studies of antifungal resistance in dermatophyte infection, which have been backed by in vitro data, thereby allowing any correlation between failure of treatment with drug resistance. Secondly, it is clear that there are strains of dermatophyte fungi that are resistant to the commonly used antifungal agents, and that investigations have been fueled by the recent spread of clinically resistant fungi, such as the T. indotineae isolates, associated with the outbreak of dermatophytosis first reported in India.

(Yellow has been deleted)

To  page 5, lines 379-384.

This study establishes two key points. Firstly, there have been very few studies of clinical antifungal resistance in dermatophyte infection, which have been backed by in vitro data, thereby allowing any correlation between failure of treatment with drug resistance. Secondly, it is clear that there are specific strains of dermatophyte fungi that are resistant to the commonly used antifungal agents, and that investigations have been fueled by the recent spread of clinically resistant fungi, such as the T. indotineae isolates, associated with the outbreak of dermatophytosis first reported in India.

  1. In Figure 1, what does wrong population, fungus and drug means?

Please find below some examples:

  • Wrong population: patients with onychomycosis or companion animals
  • Wrong fungus: articles of e.g. Fusarium spp. or Malassezia spp.
  • Wrong drugs: testing the effect of e.g. ‘rich root extracts’ or ‘Irania medical plants’ or’oil of bitter orange’
  •  

This Figure has now been changed to make the language clearer. Please see Fig 1, page 9:

Records excluded as not meeting case definition: 543 (555 reasons, some had more than one)

Onychomycosis/ tinea pedis/ tinea capitis  (n=208)

Article type or study outcome (n= 126)

Foreign language (n=62)

Population surveyed (n=52)

Fungus identified (n=43)

Drug Prescribed (n=16)

No MIC determination (n=14)

No previous treatment (n=2)

Other (n=32)

Reviewer 3 Report

The authors reviewed correlation between the results of antifungal susceptibility test and previous exposure to antifungal drugs and evaluated clinical response. The reviewer thought that they did not indicate usefulness of antifungal susceptibility test. To solve the questions, please answer the following comments.

1. In abstract, the 1st sentence and the second sentence are not linked. The purpose of this study should be clarified.

2. Line 150-151. The authors regarded Bangladesh, Iran, Japan, Mynmar, Nepal, Yemen and Sri Lanka as putative endemic countries without any evidence. This is discriminatory.

3. The authors did not clarify administration route of the previous exposure of antifungal drugs. Generally, topical administration can achieve higher concentration in the targeted area than oral administration.

4. Dermatophytoses are common disease. In many cases, the patients are treated by self-medication or small clinics using topical medicines. Do the authors claim that all dermatophytoses cases should be treated by large hospitals having a laboratory?

5. Resistance to the antifungal drug is one possible cause of failure of treatment. However, there were the other possible causes such as non-compliance, side-effect and co-morbidities, intrinsic low susceptibility. How do the authors position susceptibility test?       

Author Response

The authors reviewed correlation between the results of antifungal susceptibility test and previous exposure to antifungal drugs and evaluated clinical response. The reviewer thought that they did not indicate usefulness of antifungal susceptibility test. To solve the questions, please answer the following comments.

  1. In abstract, the 1stsentence and the second sentence are not linked. The purpose of this study should be clarified.

 We have addressed this by making the connection between antifungal treatment and MICs clearer with the second sentence which introduces the purpose of the review.

It now reads as follows:

“An increase in the number of recurrent and recalcitrant dermatophytoses calls for a tool to guide the clinician to correlate in vitro minimum inhibitory concentration (MIC) data, antifungal treatment with clinical outcomes. This systematic review aims to explore a possible correlation between one aspect of this, previous antifungal exposure, and clinical outcomes”

  1. Line 150-151. The authors regarded Bangladesh, Iran, Japan, Myanmar, Nepal, Yemen and Sri Lanka as putative endemic countries without any evidence. This is discriminatory.

Many thanks for making this point. Lines have been changed to the following (Please see lines 220 to 225):

“A total of 28% (n= 21/75) of the patients originated from or had traveled to other countries which included Bangladesh (n = 9/21), Iran (n= 6/21), Japan (n= 2/21), Myanmar (n= 1/21), Nepal (n= 1/21), Yemen (n= 1/21) and Sri Lanka (n=1/21). Some of these countries have been considered to be potentially affected in published reports of the growing spread of recalcitrant tinea infections because clinical problems with treatment failure.”

  1. The authors did not clarify administration route of the previous exposure of antifungal drugs. Generally, topical administration can achieve higher concentration in the targeted area than oral administration.

 Thank you. We have clarified this by making clear that the case definition of recalcitrant infections  specifies the use of oral therapy (please see page 3, line 206)

  1. Dermatophytoses are common disease. In many cases, the patients are treated by self-medication or small clinics using topical medicines. Do the authors claim that all dermatophytoses cases should be treated by large hospitals having a laboratory?

 Thank you. Now we have made it clear in this version that this is only applicable where facilities are available. See page 5, line 397 – 401.

”The finding of in vitro resistance, now associated with outbreaks of infection, raise concerns about the ease with which resistant strains can spread in and across community boundaries and whether MIC determinations should be more widely adopted as a standard clinical practice in dermatology where this is feasible.”

  1. Resistance to the antifungal drug is one possible cause of failure of treatment. However, there were the other possible causes such as non-compliance, side-effect and co-morbidities, intrinsic low susceptibility. How do the authors position susceptibility test?       

 Many thanks for pointing this out. The authors are in agreement and this is mentioned and referenced  in lines 58-60 in the introduction (ref 5):

“There are a number of reasons that may explain treatment failure including poor compliance with treatment, variability in the quality of antifungal drugs including generic products, misuse of strong topical corticosteroids or steroid combinations and host immune dysfunction”.

Round 2

Reviewer 2 Report

Aurthors have addressed all the comments and modified the manuscript significantly. 

Reviewer 3 Report

The revised form is well written and well answered to the reviewer's question. This manuscript should be accepted as is.